# Importance of Fluorine in Benzazole Compounds

**DOI:** 10.3390/molecules25204677

**Published:** 2020-10-14

**Authors:** Thuraya Al-Harthy, Wajdi Zoghaib, Raid Abdel-Jalil

**Affiliations:** 1Department of Basic Sciences, College of Applied and Health Sciences, A’Sharqiyah University, Ibra 400, Oman; 2Chemistry Department, College of Science, Sultan Qaboos University, Al-Khod, Muscat 123, Oman; zoghaibw@squ.edu.om

**Keywords:** benzazoles, benzimidazoles, benzothiazoles, benzoxazole, fluorine, biological activities

## Abstract

Fluorine-containing heterocycles continue to receive considerable attention due to their unique properties. In medicinal chemistry, the incorporation of fluorine in small molecules imparts a significant enhancement their biological activities compared to non-fluorinated molecules. In this short review, we will highlight the importance of incorporating fluorine as a basic appendage in benzothiazole and benzimidazole skeletons. The chemistry and pharmacological activities of heterocycles containing fluorine during the past years are compiled and discussed.

## 1. Introduction

Fluorine is one of the halogens located in the main group 7A of the periodic table of elements and it is the most electronegative element, with a Pauling assigned electronegativity value of χ = 4.0. The electronegativity leads to more polarization of C-F bond (μ C−F = 1.41 D), less covalent and more electrostatic character. Usually C–F bond does interact with its environment via electrostatic/dipole interaction. Fluorine has the smallest atomic radius among the Period 2 elements. Due to the relatively small size of fluorine compared to the hydrogen atom, it can closely mimic hydrogen in non-fluorinated analogues allowing a fluorinated precursor to fit in a given enzyme receptor and hence giving similar or enhanced biological activity. Although the carbon-fluorine bond is stronger (105.4 kcal mol^−1^) than a carbon-hydrogen bond (98.8 kcal mol^−1^) providing a higher thermal stability, yet fluorine is still a better leaving group than hydrogen. This review summarizes the literature data published in the last two decades and dealing with the biological impact of introducing fluorine as a major appendage of benzazoles. The biological activity of the most important representatives of this class of compounds are discussed.

### 1.1. Heterocycles in Medicinal Chemistry

Heterocycles are indispensable constituent molecules widely distributed in Nature and involved in many essential biomolecules. Heterocyclic organic compounds are involved in an exceptionally broad domain of reactions in Nature and their utility extends to different disciplines: like medicine [1], agriculture [2], industry and technology [3]. Fluorine was synthetically incorporated in many heterocycles in the 1970s and onwards when an appreciation of the significance of fluorine started to emerge in medicinal chemistry. Fluorinated compounds comprise about a quarter of commercially available drugs in the market [4]. The number of compounds containing fluorine has increased year by year during the past 50 years and the reputation of fluorine continues till today where among the 48 new approved drugs in 2019, there were 13-fluorine containing drugs. That is due to the unique features of fluorine, which are discussed later in details. In medicinal chemistry, since the journey of drug development is long, challenging and risky, choosing the proper moiety is highly crucial and fluorine has become a popular moiety used in small molecules to affect the biological activity. 

### 1.2. The Role of Fluorine in Medicinal Chemistry 

The following are general properties of fluorine which are exploited in medicinal chemistry and drug discovery in particular: pKa, steric effects, lipophilicity, inductive effect, hydrogen bonding, and isosterism that will be discussed later. 

Typically, fluorine substitution lowers the pKa of the nearby functionality in which protic groups become more acidic and basic groups become less basic. The pKa of an ionizable center of a drug molecule can alter its lipophilicity. The pKa can be tuned and this can manifest in the potency, selectivity, toxicity, as well as pharmacokinetic (PK) properties, which is things are manifested in absorption, distribution, metabolism, and excretion (ADME). Besides that, it can also play an important role in reducing the potential toxicity. Thus, understanding the effect of fluorine substitution on the lead compound facilitates the journey of development of therapeutic agents. In acyclic aliphatic amines, as the fluorine substitution increases, the pKa decreases. Generally, the length of the chain does not shift the pKa of the amine [5].

An attempt has been done by Fjelbye et al. to elucidate the effect of replacing a hydrogen atom with fluorine on the pKa and P-glycoprotein (Pgp)-mediated efflux for a series of phosphodiesterase type 9 (PDE9) inhibitors [6]. A harmonic trend has been noted in the pKa shift for the synthesized compounds **1** (Figure 1) after hydrogen–fluorine replacement. Consequently, these possibilities or limitations which are afforded by fluorine can be beneficial in drug discovery.

The aqueous solubility of a compound is essential in drug development whereby poor solubility affected the ADME sequence starting from the first step; poor solubility that lead to poor absorption even if the permeation rate is high. Poor solubility can be improved by increased fluorine substitution. 

Lou′s research team [7] prepared the lead compound **2** which exhibits a selective Bruton’s tyrosine kinase (BTK) inhibition with reasonable biological activity with human whole blood (HWB) IC50 of 100 nM and poor aqueous solubility (0.6/2/3 μg/mL in water). They aimed to improve the aqueous solubility, which in turn increases the biological activity which is expected to lower the effective dose. They utilized the approach that replacement of C-H bond with a C-F bond is believed to increase the metabolic stability and improve membrane permeability of a particular compound. In addition, the incorporation of fluorine in small molecules imparts higher binding affinity. In another study [7], Lou et al. demonstrated that a rational fluorine scan can be utilized to measure the potency of a series of selective BTK inhibitors. The placement of fluorine at the optimal position can lead to favorable interactions with protein side chains. Several analogues a, b and c, fluorinated and non-fluorinated at different sites in compound **2** (Figure 2); were used to investigate the fluorine moiety that could result in favorable interactions with the BTK protein. A single modification in compounds **3** and **4** increases the potency up to 40- and 20-fold when compared with non-fluorinated analogues. They concluded from their investigation that the favorable interactions and impact are attributed uniquely to the fluorine moiety.

The fluorine influence on pKa has been studied systematically in order to improve pharmaceutical properties of a thrombin inhibitor by lowering the pKa. Upon fluorine substitution, the pKa follows the expected trend and the binding affinity against thrombin and trypsin (Ki) decreases as the pKa decreases. Utilizing the linear free energy relationships (LFERs) to find out the relationship between the binding affinity and pKa, it was revealed that the thrombin inhibition is more sensitive to pKa of the amidine functionality than the inhibition of trypsin [8].

Because of the favorable half-life of the fluorine isotope ^18^F (109.8 min), it stands out as a unique atom which is used in positron emission tomography (PET). ^18^F became a powerful tool used for detection of various cancer types [9,10,11]. Ojima has introduced fluorine into taxoid anticancer agents and studied their tumor-targeted drug delivery systems [12]. The 3′-difluorovinyltaxoids (**5**) possess (Figure 3) an exceptional potency against human breast, ovarian, colon and pancreatic cancer cell lines three times higher in potency compared to that of paclitaxel against breast cancer cell lines MCF-7 (drug-sensitive) and a multidrug-resistant cell line NCI/ADR (drug-resistant) cancer cell lines respectively. 

One of the most critical concerns in drug development is the metabolism that results in reduced potency, poor PK properties or bio-activation into reactive species. An effort was done by Wan et al. to develop a series of piperidine-based 11β-hydroxysteroid dehydrogenase type I (11β-HSD1) inhibitors that may be useful targets for diabetes and obesity treatment [13]. Their work focused on improving the metabolic stability of 11β-HSD1 inhibitors. Comparing between the two compounds, the fluorine substituted **6** and the unsubstituted piperidine **7** (Figure 4) the substitution of fluorine into compound **6** has a tremendous impact metabolic stability. Due to fluorine substitution, the mouse liver microsome half-life (MLM) increased by up to 5-fold. 

In drug design, the most crucial properties that control membrane binding are the lipophilicity and the cross-sectional area. Thereby, fluorine is a good choice as a lipophilic moiety in modulating lipophilicity for bioactive compounds. Lipophilicity is often expressed as log P, the logarithm of the partition coefficient of a compound between octanol and water. It is inaccurate to generalize the fact that fluorine substitution can always increase the lipophilicity of a molecule. Prediction of fluorine substitution and its effect on lipophilicity is not straightforward since it depends on the scaffold and proximal functionality. Monofluorination or trifluorination of saturated alkyl groups often decreases lipophilicity unlike in difluorination [14].

It is worth mentioning that the emergence of fluoroquinolones in the 1980s, brought fluorine to the forefront in medicinal chemistry. Fluoroquinolones are classified as second generation quinolones and have made a paradigm shift in scientific and clinical considerations, compounds such as norfloxacin, lomefloxacin, enoxacin, ofloxacin, and ciprofloxacin are good examples [15]. Notwithstanding, fluoroquinolones have a wide spectrum of antimicrobial activity. Ciprofloxacin (**8**) (Figure 5) is a stellar example of a fluoroquinolone to date with a broad spectrum of biological activities. The incorporation of fluorine at postion-6 resulted in enhancing Gram-positive activities compared to other analogues. Fluorine plays a vital role in increasing the antibacterial potency [16], improving DNA gyrase complex binding and cell penetration. 

## 2. Benzazoles

Benzo-fused heterocycles e.g., benzazoles, are known as privileged structures which have a wide range of biological activities and are very common as lead compounds and natural products (Figure 6).

The benzazole scaffold is widely distributed in nature, e.g., benzothiazole (**9**), benzimidazole (**10**), and benzoxazole (**11**) are considered essential core structures of many biomolecules such as in firefly luciferin (**12**), a benzothiazolyl derivative exhibiting bioluminescence [17] and a benzimidazole moiety exists as an axial ligand for cobalt in vitamin B12 (**13**) [18] (Figure 1).

Although benzazole is an aromatic system; however, its skeleton contains many active sites. These active sites provide several sites for further modifications and variance at positions-2, 4, 5, 6 or 7 and the most potently active benzazoles reported are those substituted at positions 2, 5, and 6 [19,20]. 

Benzimidazole is a popular nucleus in structural manipulation in medicinal chemistry and has been intensively discussed in the literature [19,21,22,23,24] and many benzimidazoles are reported to show anthelmintic [25,26], antimicrobial [27,28,29,30,31,32], anti-HIV [33,34,35], antiviral [36,37], anti-inflammatory [38,39], anticancer [40,41,42,43,44], anticonvulsant [45], antidiabetic [45,46], antihypertensive [47], anti-tuberculosis [48,49], antimalarial [50,51] and antioxidant [52,53] activities.

Moreover, the benzothiazole core is a widespread scaffold that has been extensively studied [54,55,56,57,58,59] and many derivatives are found to be antimicrobial [49,60,61,62,63,64,65,66,67,68,69,70,71,72,73,74], anticancer [75,76,77,78,79,80,81,82,83,84,85,86,87,88,89,90,91], anti-Alzheimer’s [92,93], anti-inflammatory [94,95,96], anti-HIV [97,98,99] anticonvulsant [100,101,102], antioxidant [39,103], anti-tuberculosis [104,105], antidiabetic [106,107], antidepressant [108], antihypertension [109,110] and antimalarial [111] agents.

In addition, benzoxazole derivatives possess a variety of biological activities such as antitumor [112,113,114,115], antituberculosis [116], antifungal [117], antimicrobial [118,119,120,121,122,123], antihypertension [124], antiviral [97,125], antihormonal [126], and radioligand agents [127,128,129]. Because of the aforementioned properties of both fluorine and benzazoles, efforts have been done by Nosova et al. to summarize several methods of synthesizing fluorinated benzazoles [130]. In the next sections, each type of benzoxazole incorporated with fluorine will be discussed expressing fluorine substitution effect on its biological activity.

### 2.1. Benzothiazoles Containing Fluorine

Fluorine stands out as a distinguishable moiety in medicinal chemistry and it has been incorporated in many heterocycles, e.g., benzothiazole to improve, enhance or get new biological activity. Benzothiazole containing fluorine draws a remarkable attention in drug design since benzothiazole is considered a privileged structure and its synthesis has been discussed in many literatures. Herein are several selected examples of benzothiazole analogues (Figure 7) that manifest the importance of merging fluorine on their biological activities.

The presence of an electron-withdrawing group in a molecule e.g., fluorine increases the cytotoxic activity which was supported by several studies. For example, Li et al. prepared a new series of antitumor agents by the combination of different pharmacophores of 2-substituted-3-sulfonyl aminobenzamide with benzothiazole, orthiazolo[5,4-b]pyridine, or [1,2,4]triazolo-[1,5-a]pyridine in an attempt to enhance the potency. As a result, nineteen structures were prepared, characterized and evaluated via MTT assay against four human cancer cell lines including HCT-116, A549, MCF-7 and U-87 MG. Among the prepared compounds, fluorinated compound **14** displays potent activities against the four cancer cell lines and its anticancer effect was further tested in the nude mouse HCT-116 colon adenocarcinoma xenograft model in which BEZ235 (30 mg/kg) was used as positive drug. From the experiment, they noticed a weight loss of tested animal while using BEZ235, unlike in compound **14** and that can be referred to its low toxicity [77].

In addition, Gill et al. developed a new series of N-alkylbromo-benzothiazoles with improved efficacy and selective action. Their findings [131] suggest that the benzothiazole bearing a strong electron-withdrawing atom such as fluorine at position-6, improves the cytotoxicity against particular cancer cell lines. Fluorine benzothiazole **15** exhibits the highest potency against leukemia THP-1 cancer cell line (IC_50_ = 1 and 0.9 μM) and was higher than that of mitomycin-C (IC_50_ = 1.5 μM) when compared to other non-fluorinated derivatives. Thus, this compound can be studied further to serve as a lead compound for anticancer drug development.

In another QSAR study undertaken by Kumbhare et al., a novel class of isoxazoles and triazoles linked 2-phenyl benzothiazoles were synthesized and tested against three cancer cell lines for their anticancer activity. As a result of this study, the introduction of fluorine, in the form of (–CF_3_) group, enhanced the benzothiazole derivative **16** cytotoxicity against Colo-205 and A549 cells by causing an increase in the active caspase-3 and PARP and degradation of procaspase-8 and 9 proteins [132].

A novel series of aminobenzothiazole linked to pyrazolo[1,5-a]pyrimidine conjugates was synthesized by Kamal et al. and evaluated for their antitumor activities against five human cancer cell lines. The in vitro screening results disclosed that the 4-fluoro substituted derivatives exhibit a potent and selective anticancer activity with IC_50_ value of 1.94–3.46 μM [133]. It has been subjected to more detailed biological studies such as cell cycle analysis, tubulin polymerization studies, and Immunohistochemistry studies on tubulin. Out of these studies, compound **17** showed G2/M cell cycle arrest and 65.9% inhibition of tubulin assembly in MCF-7 cells, which makes it a promising candidate for development of breast cancer therapy.

Noteworthy, Singh et al.’s finding [62] supported the notion that the presence of electron-withdrawing groups such as fluorine, chlorine, or a combination of both, significantly affects the antibacterial activity. The systematic SAR done by the same research group showed that fluorinated variant **18** containing two di-fluoro substituents offers the highest antibacterial activity against the six tested bacterial strains (both Gram-positive and gram-negative strains) except S. boydii strains. Moreover, the modification of substituents in the benzene ring also plays a vital role in their activity showing that the position of substituents at the benzene ring is critical in governing the antibacterial activity and fluorine has a potent effect in comparison to chlorine. For instance, placing fluoro group at positions-2 and 4 in the benzene ring enhance the potency unlike position-3 where activity is lost.

Spadaro et al. synthesized a series of benzothiazole derivatives targeting inhibition of 17β-HSD1 which is a novel target for estrogen-dependent diseases [134]. As they introduced different substituents with different electronic, lipophilic, steric, and H-bonding properties, they came out with two new lead compounds. The one in the amide series **19** shows that the difluoro-derivative in the benzoyl moiety increases the inhibitory activity of the compound to reach (IC_50_ < 5 nM). This can be explained due to many factors; first, the favorable electronic effects applied by the fluorine on the H-bonding property of the OH, second, to the increased lipophilicity, or to the change in π-interacting properties of the aromatic ring.

Feng et al. prepared six cyanine dye substituted benzoxazolyl or benzothiazolyl groups and investigated them for their in vitro antiprotozoal activities. The in vitro tests were done using *P. falciparum (K1*), *Trypanosoma cruzi*, *T. brucei rhodesiense*, and *L. donovani* and the cytotoxicity of the compounds was evaluated using rat skeletal myoblasts (L-6 cells). The high antimalarial activity of the four fluorinated benzothiazolyl trimethine cyanines **20**, **21**, **22**, **23** was attributed directly to the presence of the fluorine atom [135]. Compound **20** had a high activity against *T. cruzi*, IC_50_ of 0.008 μM and a selectivity index of 193.7 and is used for Chagas disease treatment.

As mentioned previously, fluorine is a popular element which has been used as a labeling agent in the Positron Emission Tomography (PET) field [136]. Fluorine-18 labeled analogues are known as PET radio-ligands to detect and quantify the amyloid deposit distribution in human brains which is used for the detection of the development of Alzheimer’s Disease. Neumaier et al. reported the synthesis of three different [^18^F] fluoroethoxy-substituted benzothiazole derivatives. Among several PET markers prepared, [^18^F]_2_-[4′-(methylamino)phenyl]-6-fluoroethoxyben-zothiazole, (**24**) appeared as an appropriate amyloid imaging agent in terms of biokinetics and binding affinity for amyloid plaques. Its lipophilicity, (log P_oct_) was 2.4 which agrees with the requirements for amyloid imaging agents such as PIB, SB13 and FDDNP so it can penetrate the blood brain barrier [137]. The in vitro binding affinity of **24** exhibits Ki = 7.2 nM, which falls within the expected range for specific Aβ binding.

A new series of benzothiazole Schiff base derivatives was prepared by Al-Harthy et al. These derivatives were synthesized based on the most potent antibiotic fluoroquinolone, ciprofloxacin, aiming to mimic its biological activity by bearing both fluorine and piperazine moieties. The fluoroquinolone directly inhibits bacterial DNA synthesis, resulting in cell death. The fluoroquinolone mechanism of action is believed to involve inhibition of DNA gyrase and topoisomerase IV enzymes, required for supercoiling replication and separation of bacterial DNA [138]. Fluorine plays a role in controlling DNA gyrase and bacterial potency. Amongst all, the derivative that contains fluorine (**25**) shows a selective antitumor activity against DMS-53 human lung cancer cell line in comparison to primary HLMVECs [139].

Another manuscript was published later by the same researchers based on the above mentioned principle and it was based on a novel series of 5-fluoro-6-(4-methylpiperazin-1-yl)-substitutedphenylbenzo[d]thiazoles. The novel class was reported and screened against different bacteria and fungi strains. The molecule bearing electron withdrawing moiety by induction e.g., fluoro in compound **26** has a role in enhancing antibacterial activity in comparison to the non-fluorinated derivatives [140].

### 2.2. Benzimidazoles Containing Fluorine

Similar to benzothiazole, benzimidazole is known as a privileged structure involved in many therapeutic agents and its derivatives were extensively mentioned in literature especially those with a fluorine moiety. Hereinafter, are some examples of benzimidazoles (Figure 8) containing fluorine to highlight its influence on their biological activities.

To start with, Tonelli et al. prepared a set of benzimidazole derivatives as part of their continuous work in the chemistry and biological properties of benzimidazoles. In this study, a variety of moieties have been substituted at position-1 and 2 of benzimidazole; unsubstituted, basic, non-basic, aromatic, non-aromatic substitutions were synthetically introduced [37]. In order to highlight the importance of the presence of fluorine moiety, the derivative of benzimidazole with trifluoromethyl at position-2, compound **27** exhibited higher potency against CVB-5, RSV and YFV, with EC_50_ of 11, 22, and 33 μM respectively compared to its analogue **28** having a 2-methyl moiety with EC_50_ of 47, >100, and 33 μM respectively. The importance of trifluoromethyl has been proven by the narrower spectrum of activity and lower potency compared to the 2-methyl benzimidazole analogue **28**.

Zhang et al. synthesized a novel series of benzimidazole tertiary amine type of fluconazole analogues of potential antifungal activity. Fluconazole is an antifungal medicine used to treat infections caused by *Candida albicans* and *Cryptococcus neoformans*. The benzimidazole scaffold has been decorated with several moieties in order to achieve the optimal activity. For the purpose of improving the pharmacological properties, the phenyl has been substituted with chloro, fluoro, trifluoromethyl, methyl and nitro moieties into target compounds which are believed to enhance the rate of absorption and in vivo drug transportation. It is worth noting that halo-benzimidazole possesses better activity in strain growth inhibition compared to alkyl benzimidazole. Compound **29** containing a 2,4-fluorinated benzyl ring shows the highest antifungal activity against *C. albicans, S*. *cerevisiae* and *A. flavus* fungi with MIC = 16–32 mg/mL [28].

Notably, a new series of benzimidazole analogues was designed by Reddy et al. by combining benzimidazole with other heterocycles such as pyrazole in what is called hybrid molecules wherby this hybridization is believed to improve the biological activity of molecules. The newly synthesized compounds were evaluated against three human tumor cell lines: lung (A549), breast (MCF-7), cervical (HeLa) and against normal keratinocyte (HaCaT) cells using the 3-(4,5-dimethylthiazol-2-yl)-2,5-diphenyltetrazolium bromide (MTT) growth inhibition assay. Structure activity relationship (SAR) studies of these hybrids concluded that the compounds with mono-substituted halogen (fluorine, chlorine, and bromine) on benzimidazole e.g., compound **30** with a fluorine appendage showed potent cytotoxicity against tested cancer cell lines [141]. On the other hand, the incorporation of trifluoromethyl (CF_3_) substitution at position-6 of benzimidazole resulted in moderate to lower cytotoxic activity.

It is worthy to note that designing molecules targeting the inhibition of tubulin polymerization is a highly attractive approach in designing anticancer candidates. Kamal et al. synthesized benzimidazole-oxindole conjugates and evaluated them against human breast cancer cell line (MCF-7) by inhibiting tubulin polymerization. The conjugates with mono fluoro, difluo, or trifluoromethyl moieties show a considerable antiproliferative activities. Their finding implies that conjugate **31** with a difluoro moiety at position 3 and 5 on phenyl ring showed a significant cytotoxicity against breast cancer cell line (MCF-7) with an IC_50_ value of 1.59 μM. Molecular docking studies have been performed to investigate the action mode of this compound and it indicated efficient binding with the colchicine binding site [133].

Zawawi et al. explored β-glucuronidase inhibitors using structure based design of benzimidazole with 2,5-disubstitued-1,3,4-oxadiazoles. The investigation revealed that the aromatic side chains directly attached to the oxadiazole moiety influence the inhibitory potential of the benzimidazole derivatives. The structure activity relationship (SAR) study proposed the reliance of inhibition upon the aromatic ring residue and its derivatives. For instance, fluoro groups in compounds (**32**; *ortho*-fluoro: IC_50_ = 19.16 ± 0.62 μM), **33**; *para*-fluoro: IC_50_ = 13.14 ± 0.76 μM, **34**; *meta*-fluoro: IC_50_ = 16.12 ± 0.36 μM) displayed excellent inhibition regardless of the position of fluorine at the aromatic ring [142].

Singh et al. research team’s interest is focused on the preparation of several di and tri substituted bis-benzimidazoles which incorporate electron donating (OCH_3_) and electron withdrawing groups (F, Cl) on the phenyl ring and test their ability to induce DNA cleavage in the presence of mammalian topoisomerase I. Hoechst 33342 and Hoechst 33258 are examples of bis-benzimidazole derivatives which have strong binding affinity to DNA causing catalytic activity inhibition of many enzymes involved in DNA replication and synthesis. It is worth to notice that the first derivatives of Hoechst having di-substituted groups, having halogen atoms as substituents, at 2-position of benzimidazole were synthesized by his team. A considerable change in absorbance was noticed with a significant hyperchromicity and a red shift was observed in the binding ability of compound **35** to CT-DNA. This observation was correlated to the presence of fluorine at the phenyl ring in the compound [27]. The small size of fluorine enables it to slip between the DNA base pairs and chelate with them leading to unwinding of the DNA strands resulting in the UV absorption hyperchromicity shift.

The cytotoxicity of the targeted bis-benzimidazoles were investigated against human tumor cell lines, which are cervix carcinoma cell line (HeLa), breast carcinoma cell line (MCF7) and brain glioma cell line (U87) in comparison to Hoechst and Camptothecin as a reference compound. The study showed remarkable cytotoxicity to human cancer cell lines, IC50 values of 5.5 μM against MCF7 and (IC50; 1.5 μM) in HeLa. Upon investigation of the inhibitory activity against purified human topoisomerase I, the bis-benzimidazoles exhibited effective enzyme inhibition even at low concentration of 25 μM.

El-Abadelah et al. reported a synthesis of a set of 2-arylbenzimidazoles **36** bearing both piperazine and fluorine. Their synthesis was based on mimicking the value of incorporating both fluorine and piperazine in ciprofloxacin in order to achieve a broad antimicrobial activity. The in vitro evaluation against *E. coli. S. aureus*, *A. parasiticus* and *C. albicans* showed no significant antibacterial activity at concentrations ≤100 μM [143]. 

In continuation of the abovementioned work, in 2005 Abdel-Jalil et al. replaced the aryl at 2-position with an aromatic ferrocenyl unit which was then converted into the hydrochloride salt to be tested against four different pathogenic *Candida* species. At least two derivatives showed interesting potency in comparison to that of azole-based (miconazole and ketonazole) antifungal agents [144]. Another study on this area done by Abu-Elteen et al. who correlated the structures of the aforementioned 2-ferrocenyl-benzimidazoles with antifungal activity. The screening results showed that the three variants of compound **37** are the most potent against *C. albicans* [145].

### 2.3. Benzoxazoles Containing Fluorine

Benzoxazole is the third skeleton of benzazoles that exhibits a remarkable biological profile. Several benzoxazole derivatives discussed in the literature possess a wide range of biological activates mentioned earlier. In the next paragraph several examples explore the fluorine influence in some benzoxazole scaffolds (Figure 9).

Aiello et al. prepared a new class of fluorinated 2-ayl benzoxazoles, benzothiazoles and chromen-4-ones and evaluated their activity against MCF-7 and MDA 468 breast cancer cell lines and compared its activity to the known antitumor benzothiazole **38** [114]. Compound **38** is well known as a potent (GI_50_ < 0.1 nM) and selective in vitro antitumor agent in human cancer cell lines. The SAR study of these compounds shows that the presence of fluorine moiety is essential for the growth-inhibitory activity since the elimination of it or replacement of it with other halogens diminishes the inhibition ability. Although some benzoxazole derivatives **39, 40** showed excellent potency, theirs is lower than the antitumor potency of **38**.

Jauhari et al. introduced a new class of 2-[(arylidene)cyanomethyl]-5-halobenzoxazoles as part of their ongoing work in preparing antitumor, antiviral, and antimicrobial candidates. The anticancer activity was done against four sets of human cell lines (HEPG-2, HeLa, WiDr, MCF-7) [146]. Interestingly, compound **41** with fluorine at position-5 exhibited a significant activity against all four tested cell lines and an exceptional antifungal activity against both *A. flavus* and *A. niger*. In antibacterial evaluation against *Pseudomonas aeruginosa*, *Staphylococcus aureus*, and *Klebsiella pneumoniae*, this new class showed a remarkable activity compared to other analogues.

In their effort to obtain orally potent VLA-4 inhibitors, Setoguchi et al. disclosed the optimum replacement lipophilic moiety which is 7-fluoro-2-(1-methyl-1*H*-indol-3-yl)-1,3-benzoxazolyl group (**42**) with (*N*′-phenylureido) phenyl group. The fluorine placed at position-7 in the benzoxazole skeleton provides a potent activity with IC_50_ 4.7 and 156 nM (+3% HSA). The alteration of placing fluorine in the lipophilic moiety, made the compound easier to use with a certain serum protein as carrier, thus resulting in a high serum concentration. It also has excellent efficacy in a bronchial inflammatory model and favorable PK profile [147].

Omori et al. reported a novel benzoxazole NPY Y5 antagonist to construct an SAR study to identify a hit for a lead compound suitable for in vivo study [148]. From this SAR study, the replacement of *ortho* trifluoromethyl (compound **43**) caused a decrease in the IC_50_ compared with the *meta* & *para* position-substituted compounds **44**, **45**, respectively and on the other hand the alkoxy substituent at the *para* position (compound **46**) has improved the potency.

## 3. Conclusions

Fluorine is a very important moiety in bioactive molecules whereby a single modification can lead to a tremendous increase in biological activities. Therefore, there is an escalating interest in introducing fluorine in designing and developing bioactive molecules. This review highlighted the influence of introducing fluorine in some benzazole scaffolds on the pharmacological and biological profiles of these systems. On the basis of various literature surveys, fluorine-containing benzazoles show improved biological activities. In most of the cases discussed in this review, the enhancement in biological activity and lowering of toxicity have been correlated to the introduction of the fluorine substituent on benzazoles.

## Figures and Tables

**Figure 1 molecules-25-04677-f001:**
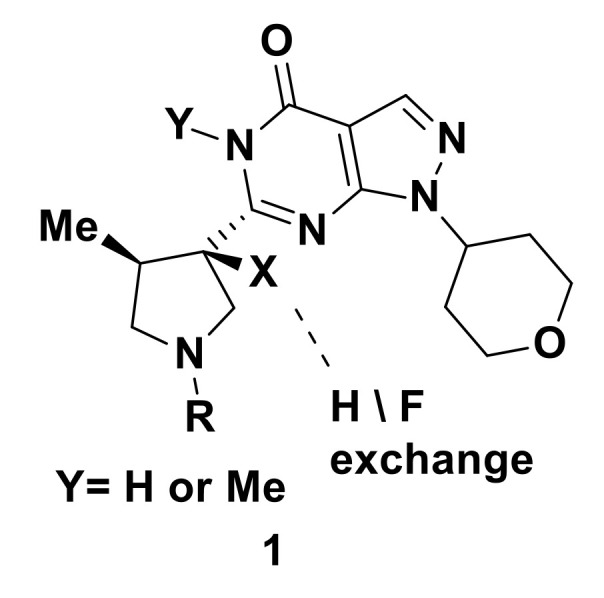
Series of PDE9 inhibitors upon H/F exchange at x position.

**Figure 2 molecules-25-04677-f002:**
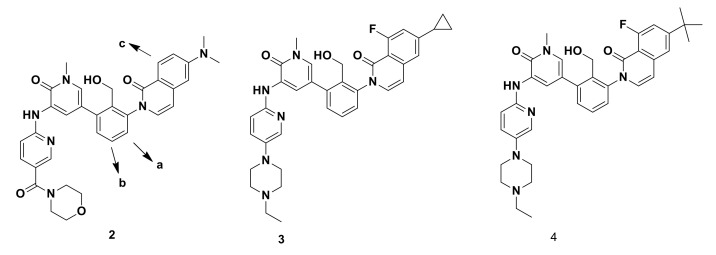
Selective BTK inhibitors.

**Figure 3 molecules-25-04677-f003:**
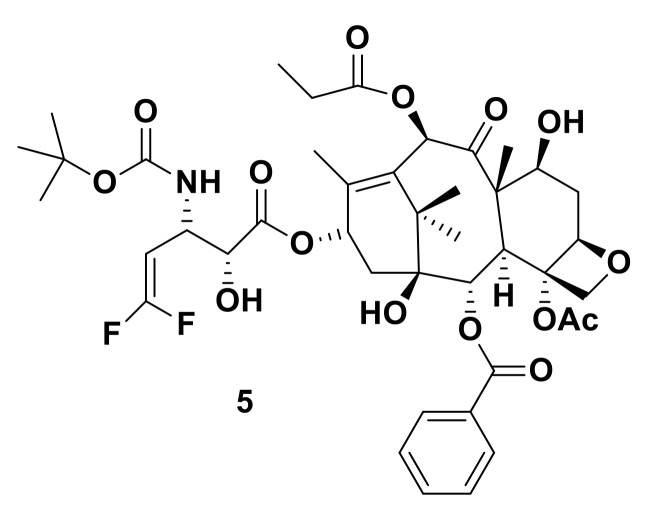
3′-Difluorovinyltaxoids.

**Figure 4 molecules-25-04677-f004:**
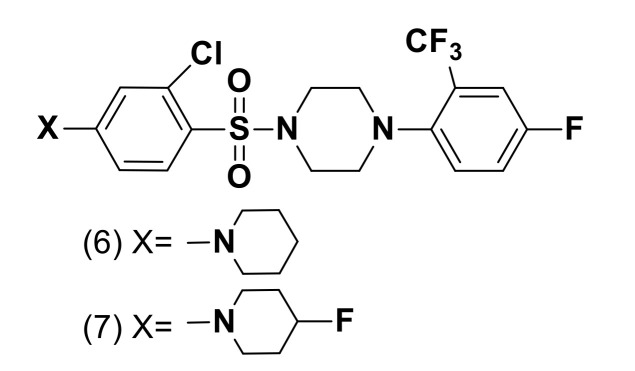
11β-HSD1 inhibitors.

**Figure 5 molecules-25-04677-f005:**
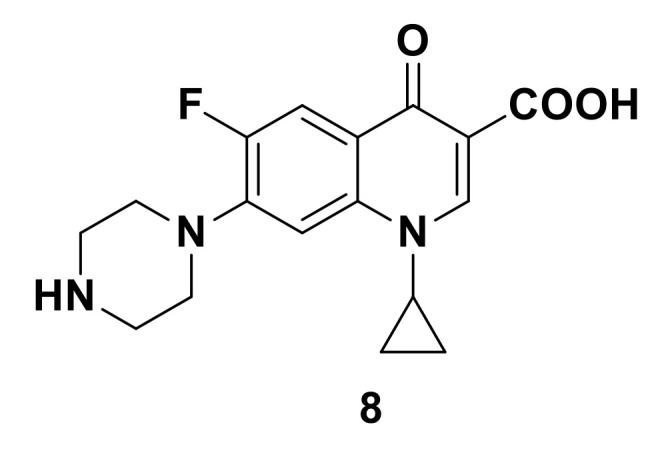
Structure of ciprofloxacin.

**Figure 6 molecules-25-04677-f006:**
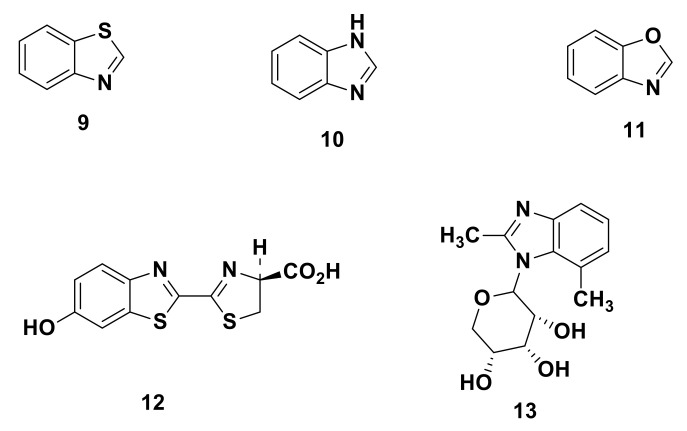
Benzazole core in some biomolecules.

**Figure 7 molecules-25-04677-f007:**
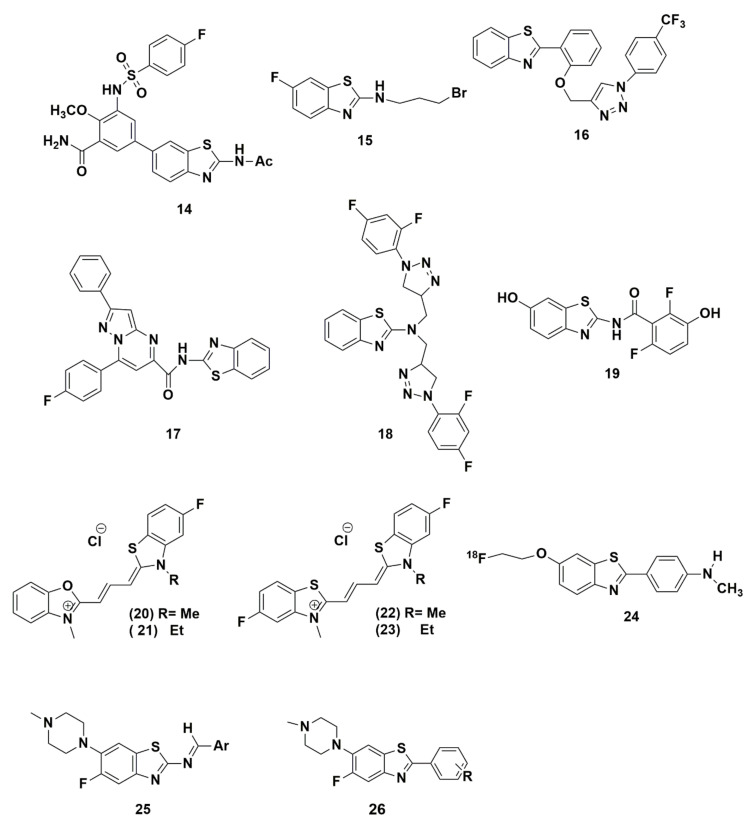
Benzothiazole derivatives incorporating fluorine.

**Figure 8 molecules-25-04677-f008:**
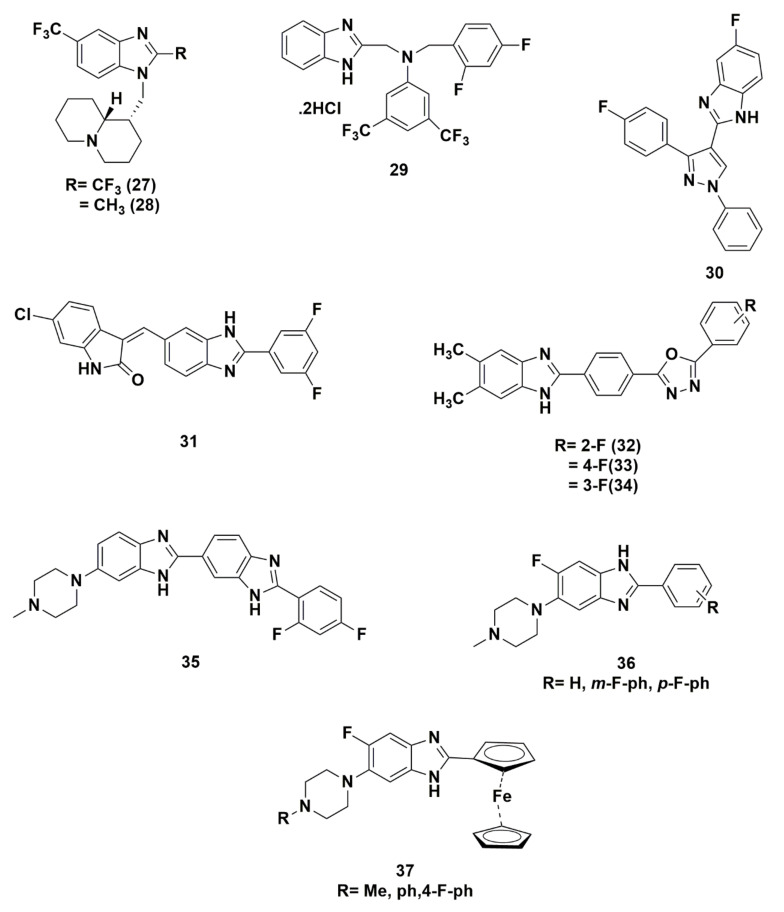
Benzimidazole derivatives incorporating fluorine.

**Figure 9 molecules-25-04677-f009:**
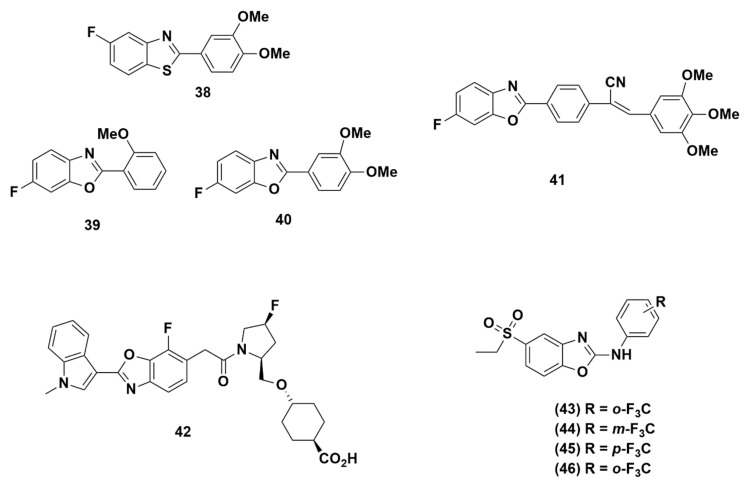
Benzoxazole derivatives incorporating fluorine.

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
