# Peer review of "Importance of Fluorine in Benzazole Compounds"

_molecules, 2020, doi:10.3390/molecules25204677_

Round 1
Reviewer 1 Report
The manuscript entitled “ Importance of Fluorine in Benzazole Compounds” by Thuraya Al-Harthy et al. describes a review on role of fluorine substituents in biologically active heterocycles, with special attention for benzothiazoles, benzimidazoles and benzoxazoles. In the initial paragraphs Authors show a general meaning of fluorine substituents in the context of „druglikeness”, i.e. ADMET/pharmacokinetics properties. The review has been performed on the basis of 145 references, predominantly coming from recent lines of evidence (2000-2019). In my opinion the review can be useful for wide group of readers representing medicinal chemistry. However, there are many editorial and linguistic errors that must be corrected before publishing in Molecules journal, according to the following comments:
- The purpose of the review should be clearly indicated in the Introduction (e.g. in the first paragraph)
- Conclusions are too general and lack of any constructive opinion in the considered research area. Authors could extend them a bit and enrich with their “expert’s point of view” useful for future studies on fluorine-containing heterocycles
- Authors used various abbreviations, which have not been explained either in the text or in any extra abbreviation list that usually is provided in the case of review article. For instance:
Line 56: “PDE9 inhibitors” – the abbreviation should be explained
Line 65: BTK – the abbreviation should be explained
Line 92 MCF7 (drug-sensitive) and NCI/ADR (drug-resistant) –should be explained
- In the chapter 1 (Introduction) all figures have been presented without title/descriptiom, ie.g.:
Line 59-60: the figure (1) should be entitled below the structure (e.g. Fig.1. Structure of…., etc.)
Line 78-79: the figure (structures 1-3) should be entitled below the structures (e.g. Fig.2. Structure of…., etc.)
- There are some minor mistakes in the data provided, e.g.:
Line 75-76 – “at different sites; a, d and c, were used” – there is no side “d” on the picture (2)
- The names of microbial strains have not been, but should be, written in italic within whole the manuscript, e.g.
line 306: E. coli. S. aureus, A. parasiticus and C. albicans
line 334-335: Pseudomonas aeruginosa, Staphylococcus aureus, and Klebsiella pneumoniae
- The manuscript is full of language mistakes and awkward phrases. I have underlined only those I found between lines 80-120, as follows:
Line 88 “F is become a powerful tool been used for detection of various cancer disease[9-10]”.
Line 88 “Ojima has introduce fluorine”
Line 90 “Theses 3′- 90 Difluorovinyltaxoids (5) possess”
Line 94 – “One of the most concern”
Line 98 - “Comparing between the two compound”
Line 100 – “mouse’s liver” and “the Mouse liver microsomes”
Line 106 – “… water. it is really inaccurate ..”
Line 111 – “Don’t forget to mention that…”
The similar disadvantages occur very frequently within the whole manuscript. Authors must carefully correct all the mistakes before publishing. I recommend that Authors send the final version of the manuscript for professional linguistic proofreading.
Author Response
Dear Reviewer 1
Please find the attached

Reviewer 2 Report
Manuscript number: molecules-878790
“Importance of Fluorine in Benzazole Compounds”
The authors present in this review their thoughts on the rationale of introducing a Fluorine group in benzazole derivatives. The idea of the work is nice and useful but the whole paper should be rewritten and better organized.
The introduction is quite confusing, without a defined structure. I would suggest to re-organize it, following a different outline: may starting from the features of fluorine and moving from the pros of introducing it in drugs. I would suggest to further divide the introduction in sub-chapters, naming them following the fluorine feature they want to highlight.
Moreover, the authors referred to the majority of the references (from 18 to 125) in a few lines, between page 4 and 5, with no additional comments than the division in therapeutic classes. I am not convinced that this is the correct way in which the references should be mentioned. If the authors consider mandatory the reference of all this works, a few sentences of them should be done. May l suggest to further divide the review not only in the three benzazoles but , for each, in the therapeutic classes they highlighted at lines 133-145. In this way, all the references should have a section and could be better defined.
Finally, a strong and complete English check should be done, may be with the help of an English native. There are too many mistakes, both in word choice or in typos, and several sentences completely lose their sense due to the difficulty in understanding it.
Author Response
Dear Reviewer 2
Please find the attached.

Reviewer 3 Report
The present manuscript would deal with both biological and pharmacological relevance of fluorinated benzazoles. Unfortunately, (1) a number of the references cited in the manuscript are not concerned to fluorinated benzazoles, and (2) the English form of the manuscript is very poor.
Since the manuscript is overwhelmed by a large number of grammar and spelling errors, the Authors should rewrite the whole text because a list of corrections would be longer than the manuscript itself.
I regret that this manuscript should not be accepted in the present form by any English-written journal.
Author Response
Dear Reviewer 3
Please find the attached

Reviewer 4 Report
This paper attempts to cover the field of fluorine-containing benzoazoles and their importance to the medicinal chemistry. In spite of the fact that those classes of compounds are of great importance for current medicinal chemistry, the referee feels that the authors did not check the all current literature. For example, references for compounds 12 and 13 are 1960 and 1961. The most important notation is that the most recent reference in the list of literature is 2018 - what about 2019 and 2020? Without information from these years the review is not comprehensive.
For the more convenient reading and understanding of the manuscript, the authors are advised to place Figures with structures right after the text in which they are discussed - not all in one picture.
Some questions and notes to the text:
- lines 56-59 and structure 1 - what is meant by "harmonic trend"? Is this for only for four compounds?
- line 88 - I would suggest adding the recent review on 18F in cathecholamine radiotracers: RUSS CHEM REV, 2018, 87, DOI:10.1070/RCR4752
- Line 153 - fluorine is not only electron-withdrawing group; it is also one of the best electron-releasing group due to its lone pairs
Minor changes:
- lines 33/34 - where is a verb?
- line 34 - "electronegativity"
- lines 34-36 - the whole phrase should be re-written
- line 37 "ha" = "has"
- line "although"
- line 57 - "pKa"
- lines 68-70 - the whole phrase is not clear
- line 95 = "pK", not "PK"
- lines 111-112 - the phrase is not clear
- line 112 - "FluoroquinoloneS IS"
- lines 122-123 - "privileged structureS" - better
- lines 130-131 - the whole phrase should be re-written
- lines 133-137 - Present Continues or Past Perfect tense
- lines 143-145 - "activities...and radioligand" - change, please
- line 147 - "Fluorine standS"
- line 149 - "Benzothiazole...drawS"
- line 157 - "prepared"
- line 160 - "against
- same line - "further"
- line 165 - "Their"
- same line - "suggest"
- line 169, 181 - "compound"
- same line - "studied"
- line 171 - "inodued" ?
- line 182 - "promising"
- line 215 - please, remove ", octanol–water partition coefficient (log Poct)" since the definition of log Poct was done earlier
- line 221 - "aiming"
- lines 228-229 - the whole phrase should be re-written
- lines 235-238 - both sentences should be re-written
- lines 243, 246- "trifluoromethyL"
- lines 251-255 - please, check English
- line 260 - "believed"
- line 263 - "StrUcture"
- et cetera...
To conclude - the text should be checked by native English-speaking person and re-written then
Author Response
Dear Reviewer 4
Please find the attached

Round 2
Reviewer 1 Report
Authors have improved the manuscript in the most important points. However, I doubt that they sent this manuscript to any professional English language correction (or the corrector was untidy).
There are still language linguistic and editorial incorrectness, e.g.:
line 32 - lack of point in the end of sentence
line 60 - fluorine substitution increases, the pKa is decreases
Compounds' signature (number) - sometimes is bolded, sometimes not (line 83 vs. 84; line 111 vs. 112)
Line 169-170 - "Herein, several selected examples of benzothiazole analogues that manifest the importance of merging fluorine on their biological activities" (the sentence without any predicate)
Line 195-196 - "A novel series of aminobenzothiazole linked to pyrazolo[1,5-a]pyrimidine conjugates were synthesized by Kamal et al."
etc.
Once Authors carefully check the manuscript and correct all such errors, the manuscript can be accepted for publication
Author Response
Dear Reviewer 1
Please find the attached.

Reviewer 2 Report
The authors reorganized the paper following the suggestions of the first revision. The review has now a better outline and has grown in scientific worth.
Nevertheless, over some minor changes, I suggest considering two important issues I noticed:
- Figures 7, 8 and 9 require to be modified, due to chemical errors, typos and badly written structures. In particular, in Fig 7 cpds 21 and 23 have Eth in their structures. I think authors refer to ethyl, which abbreviation is Et and not Eth. In Fig 8 Ph is wrongly written; in Fig 9 cpd 41 should we re-drawn and cpds 43-46 require changes in R meaning (o, m and p should be written in italic);
- The paper strongly needs an English check, absolutely by a native speaker. In addition to the huge amount of errors, the choice of the terms and words used should be better considered. Sometimes the sentences need double read to be understood.
Author Response
Dear Reviewer 2
Please find the attached

Reviewer 3 Report
No more comments are needed.
Author Response
Dear Reviewer 3
Please find the attached.

Reviewer 4 Report
Thanks to the Authors for their thoughtful work on the starting version of the manuscript.
The paper in re-written form could be published in Molecules.
Author Response
Dear Reviewer 4
Please find the attached.
